# Multi-Fidelity Active Learning with GFlowNets

## Abstract

In the last decades, the capacity to generate large amounts of data in science and engineering applications has been growing steadily. Meanwhile, the progress in machine learning has turned it into a suitable tool to process and utilise the available data. Nonetheless, many relevant scientific and engineering problems present challenges where current machine learning methods cannot yet efficiently leverage the available data and resources. For example, in scientific discovery, we are often faced with the problem of exploring very large, high-dimensional spaces, where querying a high fidelity, black-box objective function is very expensive. Progress in machine learning methods that can efficiently tackle such problems would help accelerate currently crucial areas such as drug and materials discovery. In this paper, we propose the use of GFlowNets for multi-fidelity active learning, where multiple approximations of the black-box function are available at lower fidelity and cost. GFlowNets are recently proposed methods for amortised probabilistic inference that have proven efficient for exploring large, high-dimensional spaces and can hence be practical in the multi-fidelity setting too. Here, we describe our algorithm for multi-fidelity active learning with GFlowNets and evaluate its performance in both well-studied synthetic tasks and practically relevant applications of molecular discovery. Our results show that multi-fidelity active learning with GFlowNets can efficiently leverage the availability of multiple oracles with different costs and fidelities to accelerate scientific discovery and engineering design.

## 1 Introduction

The current most pressing challenges for humanity, such as the climate crisis and the threat of pandemics or antibiotic resistance could be tackled, at least in part, with new scientific discoveries. By way of illustration, materials discovery can play an important role in improving the energy efficiency of energy production and storage; and reducing the costs and duration for drug discovery has the potential to more effectively and rapidly mitigate the consequences of new diseases. In recent years, researchers in materials science, biochemistry and other fields have increasingly adopted machine learning as a tool as it holds the promise to drastically accelerate scientific discovery [7, 67, 3, 12].

Although machine learning has already made a positive impact in scientific discovery applications [55, 27], unleashing its full potential will require improving the current algorithms [1]. For example, typical tasks in potentially impactful applications in materials and drug discovery require exploring combinatorially large, high-dimensional spaces [46, 6], where only small, noisy data sets are available, and obtaining new annotations computationally or experimentally is very expensive. Such scenarios present serious challenges even for the most advanced current machine learning methods.

In the search for a useful discovery, we typically define a quantitative proxy for usefulness, which we can view as a black-box function. One promising avenue for improvement is developing methods that more efficiently leverage the availability of multiple approximations of the target black-box

function at lower fidelity but much lower cost than the highest fidelity oracle [10, 14]. For example, the most accurate estimation of the properties of materials and molecules is only typically obtained via synthesis and characterisation in a laboratory. However, this is only feasible for a small number of promising candidates. Approximate quantum mechanics simulations of a larger amount of chemical compounds can be performed via Density Functional Theory (DFT) [41, 51]. However, DFT is still computationally too expensive for high-throughput exploration of large search spaces. Thus, large-scale exploration can only be achieved through cheaper but less accurate oracles. Nonetheless, solely relying on low-fidelity approximations is clearly suboptimal. Ideally, such tasks would be best tackled by methods that can efficiently and adaptively distribute the available computational budget between the multiple oracles depending on the already acquired information.

The past decade has seen significant progress in multi-fidelity Bayesian optimisation (BO) [19, 53], including methods that leverage the potential of deep neural networks [36]. Although highly relevant for scientific discovery, standard BO is not perfectly suited for some of the challenges in materials and drug discovery tasks. First and foremost, BO's ultimate goal is to find the optimum of an expensive black-box function. However, even the highest fidelity oracles in such problems are underspecified with respect to the actual, relevant, downstream applications. Therefore, it is imperative to develop methods that, instead of "simply" finding the optimum, discover a set of diverse, high-scoring candidates.

Recently, generative flow networks (GFlowNets) [4] have demonstrated their capacity to find diverse candidates through discrete probabilistic modelling, with particularly promising results when embedded in an active learning loop [22]. Here, we propose to extend the applicability of GFlowNets for multi-fidelity active learning.

In this paper, we present an algorithm for multi-fidelity active learning with GFlowNets. We provide empirical results in two synthetic benchmark tasks and four practically relevant tasks for biological sequence design and molecular modelling. As a main result, we demonstrate that multi-fidelity active learning with GFlowNets discovers diverse, high-scoring samples when multiple oracles with different fidelities and costs are available, with lower computational cost than its single-fidelity counterpart.

## 2 Related Work

Our work can be framed within the broad field of active learning (AL), a class of machine learning methods whose goal is to learn an efficient data sampling scheme to accelerate training [50]. For the bulk of the literature in AL, the goal is to train an accurate model $h(x)$ of an unknown target function $f(x)$, as in classical supervised learning. However, in certain scientific discovery problems, which is the motivation of our work, a desirable goal is often to discover multiple, diverse candidates $x$ with high values of $f(x)$. The reason is that the ultimate usefulness of a discovery is extremely expensive to quantify and we always rely on more or less accurate approximations. Since we generally have the option to consider more than one candidate solution, it is safer to generate a set of diverse and apparently good solutions, instead of focusing on the single global optimum of the wrong function.

This distinctive goal is closely connected to related research areas such as Bayesian optimisation [19] and active search [20]. Bayesian optimisation (BO) is an approach grounded in Bayesian inference for the problem of optimising a black-box objective function $f(x)$ that is expensive to evaluate. In contrast to the problem we address in this paper, standard BO typically considers continuous domains and works best in relatively low-dimensional spaces [18]. Nonetheless, in recent years, approaches for BO with structured data [13] and high-dimensional domains [21] have been proposed in the literature. The main difference between BO and the problem we tackle in this paper is that we are interested in finding multiple, diverse samples with high value of $f$ and not only the optimum.

This goal, as well as the discrete nature of the search space, is shared with active search, a variant of active learning in which the task is to efficiently find multiple samples of a valuable (binary) class from a discrete domain $\mathcal{X}$ [20]. This objective was already considered in the early 2000s by Warmuth et al. for drug discovery [59], and more formally analysed in later work [26, 25]. A recent branch of research in stochastic optimisation that considers diversity is so-called Quality-Diversity [9], which typically uses evolutionary algorithms that perform search in the latent space. All these and other problems such as multi-armed bandits [48] and the general framework of experimental design [8] all

share the objective of optimising or exploring an expensive black-box function. Formal connections between some of these areas have been established in the literature [54, 17, 23, 15].

Multi-fidelity methods have been proposed in most of these related areas of research. An early survey on multi-fidelity methods for Bayesian optimisation was compiled by Peherstorfer et al. [42], and research on the subject has continued since [44, 53], with the proposal of specific acquisition functions [56] and the use of deep neural networks to improve the modelling [36]. Interestingly, the literature on multi-fidelity active learning [35] is scarcer than on Bayesian optimisation. Recently, works on multi-fidelity active search have also appeared in the literature [40]. Finally, multi-fidelity methods have recently started to be applied in scientific discovery problems [10, 14]. However, the literature is still scarce probably because most approaches do not tackle the specific needs in scientific discovery, such as the need for diverse samples. Here, we aim addressing this need with the use of GFlowNets [4, 24] for multi-fidelity active learning.

# 3 Method

## 3.1 Background

**GFlowNets**   Generative Flow Networks [GFlowNets; 4, 5] are amortised samplers designed for sampling from discrete high-dimensional distributions. Given a space of compositional objects $\mathcal{X}$ and a non-negative reward function $R(x)$, GFlowNets are designed to learn a stochastic policy $\pi$ that generates $x \in \mathcal{X}$ with a probability proportional to the reward, that is $\pi(x) \propto R(x)$. This distinctive property induces sampling diverse, high-reward objects, which is a desirable property for scientific discovery, among other applications [23].

The objects $x \in \mathcal{X}$ are constructed sequentially by sampling transitions $s_t \rightarrow s_{t+1} \in \mathbb{A}$ between partially constructed objects (states) $s \in \mathcal{S}$, which includes a unique empty state $s_0$. The stochastic forward policy is typically parameterised by a neural network $P_F(s_{t+1}|s_t; \theta)$, where $\theta$ denotes the learnable parameters, which models the distribution over transitions $s_t \rightarrow s_{t+1}$ from the current state $s_t$ to the next state $s_{t+1}$. The backward transitions are parameterised too and denoted $P_B(s_t|s_{t+1}; \theta)$. The probability $\pi(x)$ of generating an object $x$ is given by $P_F$ and its sequential application:

$$\pi(x) = \sum_{\tau: s_{|\tau|-1} \rightarrow x \in \tau} \prod_{t=0}^{|\tau|-1} P_F(s_{t+1}|s_t; \theta),$$

which sums over all trajectories $\tau$ with terminating state $x$, where $\tau = (s_0 \rightarrow s_1 \ldots \rightarrow s_{|\tau|})$ is a complete trajectory. To learn the parameters $\theta$ such that $\pi(x) \propto R(x)$ we use the trajectory balance learning objective [37]

$$\mathcal{L}_{TB}(\tau; \theta) = \left( \log \frac{Z_\theta \prod_{t=0}^{n} P_F(s_{t+1}|s_t; \theta)}{R(x) \prod_{t=1}^{n} P_B(s_t|s_{t+1}; \theta)} \right)^2, \tag{1}$$

where $Z_\theta$ is an approximation of the partition function $\sum_{x \in \mathcal{X}} R(x)$ that is learned. The GFlowNet learning objective supports training from off-policy trajectories, so for training the trajectories are typically sampled from a mixture of the current policy with a uniform random policy. The reward is also tempered to make the policy focus on the modes.

**Active Learning**   In its simplest formulation, the active learning problem that we consider is as follows: we start with an initial data set $\mathcal{D} = \{(x_i, f(x_i))\}$ of samples $x \in \mathcal{X}$ and their evaluations by an expensive, black-box objective function (oracle) $f : \mathcal{X} \rightarrow \mathbb{R}$, which we use to train a surrogate model $h(x)$. A GFlowNet can then be trained to learn a generative policy $\pi_\theta(x)$ using $h(x)$ as reward function, that is $R(x) = h(x)$. Optionally, we can instead train a probabilistic proxy $p(f|\mathcal{D})$ and use as reward the output of an acquisition function $\alpha(x, p(f|\mathcal{D}))$ that considers the epistemic uncertainty of the surrogate model, as typically done in Bayesian optimisation. Finally, we use the policy $\pi(x)$ to generate a batch of samples to be evaluated by the oracle $f$, we add them to our data set and repeat the process a number of active learning rounds.

While much of the active learning literature [50] has focused on so-called *pool-based* active learning, where the learner selects samples from a pool of unlabelled data, we here consider the scenario of *de novo query synthesis*, where samples are selected from the entire object space $\mathcal{X}$. This scenario

is particularly suited for scientific discovery [30, 62, 64, 33]. The ultimate goal pursued in active learning applications is also heterogeneous. Often, the goal is the same as in classical supervised machine learning: to train an accurate (proxy) model $h(x)$ of the unknown target function $f(x)$. For some problems in scientific discovery, we are usually not interested in the accuracy in the entire input space $\mathcal{X}$, but rather in discovering new, diverse objects with high values of $f$. This is connected to other related problems such as Bayesian optimisation [19], active search [20] or experimental design [8], as reviewed in Section 2.

## 3.2 Multi-Fidelity Active Learning

We now consider the following active learning problem with multiple oracles of different fidelities. Our ultimate goal is to generate a batch of $K$ samples $x \in \mathcal{X}$ according to the following desiderata:

- The samples obtain a high value when evaluated by the objective function $f : \mathcal{X} \rightarrow \mathbb{R}^+$.

- The samples in the batch should be distinct and diverse, that is cover distinct high-valued regions of $f$.

Furthermore, we are constrained by a computational budget $\Lambda$ that limits our capacity to evaluate $f$. While $f$ is extremely expensive to evaluate, we have access to a discrete set of surrogate functions (oracles) $\{f_m\}_{1 \leq m \leq M} : \mathcal{X} \rightarrow \mathbb{R}^+$, where $m$ represents the fidelity index and each oracle has an associated cost $\lambda_m$. We assume $f_M = f$ because there may be even more accurate oracles for the true usefulness but we do not have access to them, which means that even when measured by $f = f_M$, diversity remains an important objective. We also assume, without loss of generality, that the larger $m$, the higher the fidelity and that $\lambda_1 < \lambda_2 < \ldots < \lambda_M$. This scenario resembles many practically relevant problems in scientific discovery, where the objective function $f_M$ is indicative but not a perfect proxy of the true usefulness of objects $x$—hence we want diversity—yet it is extremely expensive to evaluate—hence cheaper, approximate models are used in practice.

In multi-fidelity active learning—as well as in multi-fidelity Bayesian optimisation—the iterative sampling scheme consists of not only selecting the next object $x$ (or batch of objects) to evaluate, but also the level of fidelity $m$, such that the procedure is cost-effective.

Our algorithm, MF-GFN, detailed in Algorithm 1, proceeds as follows: An active learning round $j$ starts with a data set of annotated samples $\mathcal{D}_j = \{(x_i, f_m(x_i), m_i)\}_{1 \leq m \leq M}$. The data set is used to fit a probabilistic *multi-fidelity surrogate* model $h(x, m)$ of the posterior $p(f_m(x)|x, m, \mathcal{D})$. We use Gaussian Processes [47], as is common in Bayesian optimisation, to model the posterior, such that the model $h$ predicts the conditional Gaussian distribution of $f_m(x)$ given $(x, m)$ and the existing data set $\mathcal{D}$. We implement a multi-fidelity GP kernel by combining a Matern kernel evaluated on $x$ with a linear downsampling kernel over $m$ [61]. In the higher dimensional problems, we use Deep Kernel Learning [60] to increase the expressivity of the surrogate models. The candidate $x$ is modelled with the deep kernel while the fidelity $m$ is modelled with the same linear downsamling kernel. The output of the proxy model is then used to compute the value of a *multi-fidelity acquisition function* $\alpha(x, m)$. In our experiments, we use the multi-fidelity version [56] of Max-Value Entropy Search (MES) [58], which is an information-theoretic acquisition function widely used in Bayesian optimization. MES aims to maximize the mutual information between the value of the queried $x$ and the maximum value attained by the objective function, $f^\star$. The multi-fidelity variant is designed to select the candidate $x$ and the fidelity $m$ that maximize the mutual information between $f_M^\star$ and the oracle at fidelity $m$, $f_m$, weighted by the cost of the oracle:

$$\alpha(x, m) = \frac{1}{\lambda_m} I(f_M^\star; f_m | \mathcal{D}_j). \tag{2}$$

We provide further details about the acquisition function in Appendix A. A multi-fidelity acquisition function can be regarded as a cost-adjusted utility function. Therefore, in order to carry out a cost-aware search, we seek to sample diverse objects with high value of the acquisition function. In this paper, we propose to use a GFlowNet as a generative model trained for this purpose (see further details below in Section 3.3). An active learning round terminates by generating $N$ objects from the sampler (here the GFlowNet policy $\pi$) and forming a batch with the best $B$ objects, according to $\alpha$. Note that $N \gg B$, since sampling from a GFlowNet is relatively inexpensive. The selected

objects are annotated by the corresponding oracles and incorporated into the data set, such that
$\mathcal{D}_{j+1} = \mathcal{D}_j \cup \{(x_1, f_m(x_1), m_1), \ldots (x_B, f_m(x_B), m_B)\}$.

---

**Algorithm 1:** MF-GFN: Multi-fidelity active learning with GFlowNets. See Section 4.1 for quality (Top-$K(\mathcal{D})$) and diversity metrics.

---

**Input:** $\{(f_m, \lambda_m)\}$: $M$ oracles and their corresponding costs;
$\mathcal{D}_0 = \{(x_i, f_m(x_i), m_i)\}$: Initial dataset;
$h(x, m)$: Multi-fidelity Gaussian Process proxy model;
$\alpha(x, m)$: Multi-fidelity acquisition function;
$R(\alpha(x), \beta)$: reward function to train the GFlowNet;
$B$: Batch size of oracles queries;
$\Lambda$: Maximum available budget;
$K$: Number of top-scoring candidates to be evaluated at termination;
**Result:** Top-$K(\mathcal{D})$, Diversity
**Initialization:** $\Lambda_j = 0$, $\mathcal{D} = \mathcal{D}_0$
**while** $\Lambda_j < \Lambda$ **do**
> • Fit $h$ on dataset $\mathcal{D}$;
> • Train GFlowNet with reward $R(\alpha(x), \beta)$ to obtain policy $\pi_\theta(x)$;
> • Sample $B$ tuples $(x_i, m_i) \sim \pi_\theta$;
> • Evaluate each tuple with the corresponding oracle to form batch
>   $\mathcal{B} = \{(x_1, f_m(x_1), m_1), \ldots, (x_B, f_m(x_B), m_B)\}$;
> • Update dataset $\mathcal{D} = \mathcal{D} \cup \mathcal{B}$;

**end**

---

### 3.3 Multi-Fidelity GFlowNets

In order to use GFlowNets in the multi-fidelity active learning loop described above, we propose to make the GFlowNet sample the fidelity $m$ for each object $x \in \mathcal{X}$ in addition to $x$ itself. Formally, given a baseline GFlowNet with state and transition spaces $\mathcal{S}$ and $\mathbb{A}$, we augment the state space with a new dimension for the fidelity $\mathcal{M}' = \{0, 1, 2, \ldots, M\}$ (including $m = 0$, which corresponds to unset fidelity), such that the augmented, multi-fidelity space is $\mathcal{S}_{\mathcal{M}'} = \mathcal{S} \cup \mathcal{M}'$. The set of allowed transitions $\mathbb{A}_M$ is augmented such that a fidelity $m > 0$ of a trajectory must be selected once, and only once, from any intermediate state.

Intuitively, allowing the selection of the fidelity at any step in the trajectory should give flexibility for better generalisation. At the end, finished trajectories are the concatenation of an object $x$ and the fidelity $m$, that is $(x, m) \in \mathcal{X}_{\mathcal{M}} = \mathcal{X} \cup \mathcal{M}$. In summary, the proposed approach enables to jointly learn the policy that samples objects in a potentially very large, high-dimensional space, together with the level of fidelity, that maximise a given multi-fidelity acquisition function as reward.

## 4 Empirical Evaluation

In this section, we describe the evaluation metrics and experiments performed to assess the validity and performance of our proposed approach of multi-fidelity active learning with GFlowNets. Overall, the purpose of this empirical evaluation is to answer the following questions:

- **Question 1**: Is our multi-fidelity active learning approach able to find high-scoring, diverse samples at lower cost than active learning with a single oracle?

- **Question 2**: Does our proposed multi-fidelity GFlowNet, which learns to sample fidelities together with objects $(x, m)$, provide any advantage over sampling only objects $x$?

In Section 4.1 we describe the metrics proposed to evaluate the performance our proposed method, as well as the baselines, which we describe in Section 4.2. In Section 4.3, we present results on synthetic tasks typically used in the multi-fidelity BO and active learning literature. In Section 4.4, we present results on more practically relevant tasks for scientific discovery, such as the design of DNA sequences and anti-microbial peptides.

## 4.1 Metrics

One core motivation in the conception of GFlowNets, as reported in the original paper [4], was the goal of sampling diverse objects with high-score, according to a reward function.

- Mean score, as per the highest fidelity oracle $f_M$, of the top-$K$ samples.
- Mean pairwise similarity within the top-$K$ samples.

Furthermore, since here we are interested in the cost effectiveness of the active learning process, in this section we will evaluate the above metrics as a function of the cost accumulated in querying the multi-fidelity oracles. It is important to note that the multi-fidelity approach is not aimed at achieving *better* mean top-$K$ scores than a single-fidelity active learning counterpart, but rather *the same* mean top-$K$ scores *with a smaller budget*.

## 4.2 Baselines

In order to evaluate our approach, and to shed light on the questions stated above, we consider the following baselines:

- **GFlowNet with highest fidelity (SF-GFN):** GFlowNet based active learning approach from [22] with the highest fidelity oracle to establish a benchmark for performance without considering the cost-accuracy trade-offs.
- **GFlowNet with random fidelities (Random fid. GFlowNet):** Variant of SF-GFN where the candidates are generated with the GFlowNet but the fidelities are picked randomly and a multi-fidelity acquisition function is used, to investigate the benefit of deciding the fidelity with GFlowNets.
- **Random candidates and fidelities (Random):** Quasi-random approach where the candidates and fidelities are picked randomly and the top $(x, m)$ pairs scored by the acquisition function are queried.
- **Multi-fidelity PPO (MF-PPO):** Instantiation of multi-fidelity Bayesian optimization where the acquisition function is optimized using proximal policy optimization [PPO 49].

## 4.3 Synthetic Tasks

As an initial assessment of MF-GFNs, we consider two synthetic functions—Branin and Hartmann—widely used in the single- and multi-fidelity Bayesian optimisation literature [44, 53, 28, 36, 16].

**Branin** We consider an active learning problem in a two-dimensional space where the target function $f_M$ is the Branin function, as modified in [52] and implemented in `botorch` [2]. We simulate three levels of fidelity, including the true function. The lower-fidelity oracles, the costs of the oracles (0.01, 0.1, 1.0) as well as the number of points queried in the initial training set were adopted from [36]. We provide further details about the task in Appendix B.2. In order to consider a discrete design space, we map the domain to a discrete $100 \times 100$ grid. We model this grid with a GFlowNet as in [4, 37]: starting from the origin $(0, 0)$, for any state $s = (x_1, x_2)$, the action space consists of the choice between the exit action or the dimension to increment by 1, provided the next state is in the limits of the grid. Fig. 1a illustrates the results for this task. We observe that MF-GFN is able to reach the minimum of the Branin function with a smaller budget than the single-fidelity counterpart and the baselines.

**Hartmann** Next, we consider the 6-dimensional Hartmann function as objective $f_M$ on a hyper-grid domain. As with Branin, we consider three oracles, adopting the lower-fidelity oracles and the set of costs (0.125, 0.25, 1.0) from [53]. We discretize the domain into a six-dimensional hyper-grid of length 10, yielding $10^6$ possible candidate points. The results for the task are illustrated in Fig. 1b, which indicate that multi-fidelity active learning with GFlowNets (MF-GFN) offers an advantage over single-fidelity active learning (SF-GFN) as well as some of the other baselines in this higher-dimensional synthetic problem as well. Note that while MF-PPO performs better in this task, as shown in the next experiments, MF-PPO tends to collapse to single modes of the function in more complex high-dimensional scenarios.

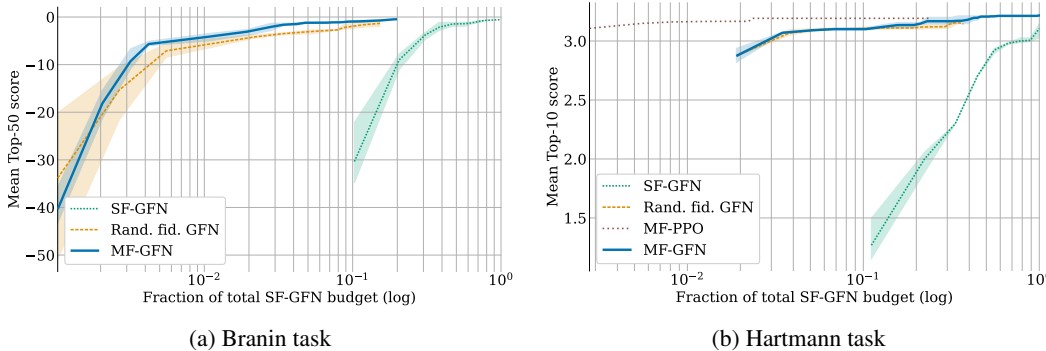

(a) Branin task                              (b) Hartmann task

Figure 1: Results on the synthetic tasks—Branin and Hartmann functions. The curves indicate the mean score $f_M$ within the top-50 and top-10 samples (for Branin and Hartmann, respectively) computed at the end of each active learning round and plotted as a function of the budget used. The random baseline is omitted from this plot to facilitate the visualisation since the results were significantly worse in these tasks. We observe that MF-GFN clearly outperforms the single-fidelity counterpart (SF-GFN) and slightly improves upon the GFlowNet baseline that samples random fidelities. On Hartmann, MF-PPO initially outperforms the other methods.

## 4.4 Benchmark Tasks

While the synthetic tasks are insightful and convenient for analysis, to obtain a more solid assessment of the performance of MF-GFN, we evaluate it, together with the other baselines, on more complex, structured design spaces of practical relevance. We present results on a variety of tasks including DNA aptamers (Section 4.4.1), anti-microbial peptides (Section 4.4.2) and small molecules (Section 4.4.3).

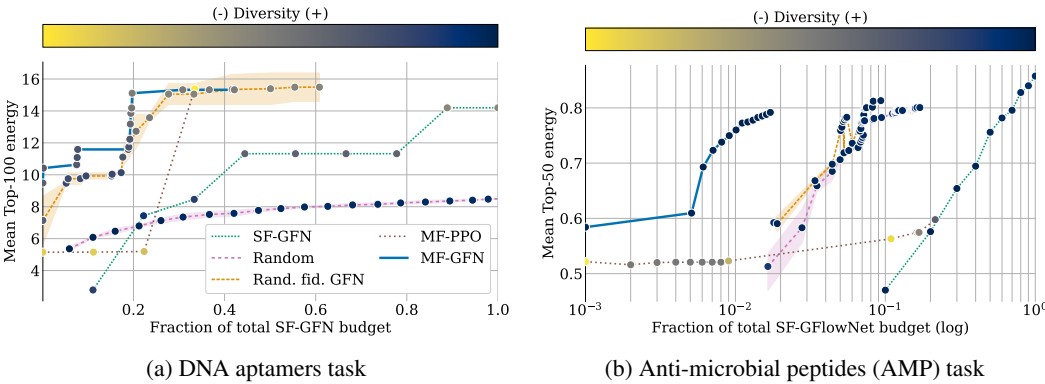

(a) DNA aptamers task                   (b) Anti-microbial peptides (AMP) task

Figure 2: Results on the DNA aptamers and AMP tasks. The curves indicate the mean score $f_M$ within the top-100 and top-50 samples (for DNA and AMP, respectively) computed at the end of each active learning round and plotted as a function of the budget used. The colour of the markers indicates the diversity within the batch (darker colour of the circular dots indicating more diversity). In both the DNA and AMP tasks, MF-GFN outperforms all baselines in terms of cost efficiency, while obtaining great diversity in the final batch of top-$K$ candidates.

## 4.4.1 DNA Aptamers

DNA aptamers are single-stranded nucleotide sequences with multiple applications in polymer design due to their specificity and affinity as sensors in crowded biochemical environments [66, 11, 63, 29]. DNA sequences are represented as strings of nucleobases A, C, T or G. In our experiments, we consider fixed-length sequences of 30 bases and design a GFlowNet environment where the action space $\mathbb{A}$ consists of the choice of base to append to the sequence, starting from an empty sequence. This yields a design space of size $|\mathcal{X}| = 4^{30}$ (ignoring the selection of fidelity in MF-GFN). As the optimisation objective $f_M$ (highest fidelity) we used the free energy of the secondary structure as calculated by NUPACK [65]. As a lower fidelity oracle, we trained a transformer model on 1

million randomly sampled sequences annotated with $f_M$, and assigned it a cost $100\times$ smaller than the highest-fidelity oracle. Further details about the task are discussed in Appendix B.4.

The main results on the DNA aptamers task are presented in Fig. 2a. We observe that on this task MF-GFN outperforms all other baselines in terms cost efficiency. For instance, MF-GFN achieves the best mean top-$K$ energy achieved by its single-fidelity counterpart with just about $20\%$ of the budget. It is also more efficient than GFlowNet with random fidelities and MF-PPO. Crucially, we also see that MF-GFN maintains a high level of diversity, even after converging to topK reward. On the contrary, MF-PPO is not able to discover diverse samples, as is expected based on prior work [22].

### 4.4.2 Antimicrobial Peptides

Antimicrobial peptides are short protein sequences which possess antimicrobial properties. As proteins, these are sequences of amino-acids—a vocabulary of 20 along with a special stop token. We consider variable length proteins sequences with up to 50 residues. We use data from DBAASP [45] containing antimicrobial activity labels, which is split into two sets – one used for training the oracle and one as the initial dataset in the active learning loop, following [22]. To establish the multi-fidelity setting, we train different models with different capacities and with different subsets of the data. The details about these oracles along with additional details about the task are discussed in Appendix B.5.

The results in Fig. 2b inidicate that even in this task MF-GFN outperforms all other baselines in terms of cost-efficiency. It reaches the same maximum mean top-$K$ score as the random baselines with $10\times$ less budget and almost $100\times$ less budget than SF-GFN. In this task, MF-PPO did not achieve comparable results. Crucially, the diversity of the final batch found by MF-GFN stayed high, satisfying this important criterion in the motivation of this method.

### 4.4.3 Small Molecules

Molecules are clouds of interacting electrons (and nuclei) described by a set of quantum mechanical descriptions, or properties. These properties dictate their chemical behaviours and applications. Numerous approximations of these quantum mechanical properties have been developed with different methods at different fidelities, with the famous example of Jacob's ladder in density functional theory [43]. To demonstrate the capability of MF-GFlowNet to function in the setting of quantum chemistry, we consMF-GFNoof-of-concept tasks in molecular electronic potentials: maximization of adiabatic electron affinity (EA) and (negative) adiabatic ionisation potential (IP). These electronic potentials dictate the molecular redox chemistry, and are key quantities in organic semiconductors, photoredox catalysis, or organometallic synthesis. We employed three oracles that correlate with experimental results as approximations of the scoring function, by uses of varying levels of geometry optimisation to obtain approximations to the adiabatic geometries, followed by the calculation of IP or EA with semi-empirical quantum chemistry XTB (see Appendix) [39]. These three oracles had costs of 1, 3 and 7 (respectively), proportional to their computational running demands. We designed the GFlowNet state space by using sequences of SELFIES tokens [32] (maximum of 64) to represent molecules, starting from an empty sequence; every action consists of appending a new token to the sequence.

The realistic configuration and practical relevance of these tasks allow us to draw stronger conclusions about the usefulness of multi-fidelity active learning with GFlowNets in scientific discovery applications. As in the other tasks evaluated, we here also found MF-GFN to achieve better cost efficiency at finding high-score top-$K$ molecules (Fig. 3), especially for ionization potentials (Fig. 3a). By clustering the generated molecules, we find that MF-GFN captures as many modes as random generation, far exceeding that of MF-PPO. Indeed, while MF-PPO seems to outperform MF-GFN in the task of electron affinity (Fig. 3b), all generated molecules were from a few clusters, which is of much less utility for chemists.

## 5 Conclusions, Limitations and Future Work

In this paper, we present MF-GFN, the first application of GFlowNets for multi-fidelity active learning. Inspired by the encouraging results of GFlowNets in (single-fidelity) active learning for biological sequence design [22] as a method to discover diverse, high-scoring candidates, we propose MF-GFN

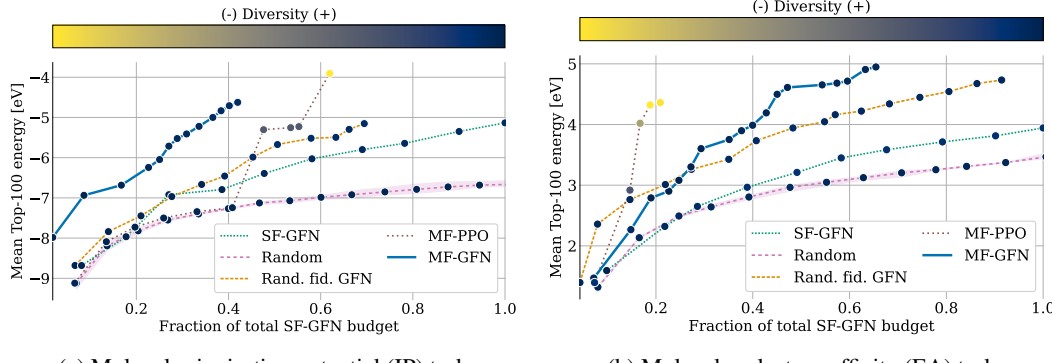

(a) Molecules ionisation potential (IP) task      (b) Molecules electron affinity (EA) task

Figure 3: Comparative results on the molecular discovery tasks: (a) ionisation potential (IP), (b) electron affinity (EA). Results illustrate the generally faster convergence of MF-GFN to discover a diverse set of molecules with desirable values of the target property (colour scheme of the circular dots: darker/blue is better than lighter/yellow).

to sample the candidates as well as the fidelity at which the candidate is to be evaluated, when multiple oracles are available with different fidelities and costs.

We evaluate the proposed MF-GFN approach in both synthetic tasks commonly used in the multi-fidelity Bayesian optimization literature and benchmark tasks of practical relevance, such as DNA aptamer generation, antimicrobial peptide design and molecular modelling. Through comparisons with previously proposed methods as well as with variants of our method designed to understand the contributions of different components, we conclude that multi-fidelity active learning with GFlowNets not only outperforms its single-fidelity active learning counterpart in terms of cost effectiveness and diversity of sampled candidates, but it also offers an advantage over other multi-fidelity methods due to its ability to learn a stochastic policy to jointly sample objects and the fidelity of the oracle to be used to evaluate them.

**Broader Impact**    Our work is motivated by pressing challenges to sustainability and public health, and we envision applications of our approach to drug discovery and materials discovery. However, as with all work on these topics, there is a potential risk of dual use of the technology by nefarious actors [57].

**Limitations and Future Work**    Aside from the molecular modelling tasks, our empirical evaluations in this paper involved simulated oracles with relatively arbitrary costs. Therefore, future work should evaluate MF-GFN with practical oracles and sets of costs that reflect their computational or financial demands. Furthermore, we believe a promising avenue that we have not explored in this paper is the application of MF-GFN in more complex, structured design spaces, such as hybrid (discrete and continuous) domains [34], as well as multi-fidelity, multi-objective problems [24].

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
