# OpenReview forum: "Multi-Fidelity Active Learning with GFlowNets"
_NeurIPS.cc/2023/Conference — Submitted to NeurIPS 2023_

### Official Review · Reviewer_3rWi · 2023-07-06

**Soundness:** 3 good
**Presentation:** 3 good
**Contribution:** 2 fair
**Rating:** 5
**Confidence:** 4

**Summary:**

In this manuscript, the authors propose a multi-fidelity active learning scheme based on GFlowNets.
The work mainly aims to tackle scientific discovery problems, where one often faces exploring a huge high-dimensional space to identify novel, diverse, high-quality solutions.
In many scientific applications, accurately evaluating the quality of the potential solutions (or properties/characteristics of novel candidates) is expensive, hence lower-fidelity surrogate models are frequently adopted for efficient cost-effective evaluation.
The current work investigates how to carry out active learning - more specifically, in the context of de novo query synthesis instead of a pool-based active learning scenario - when multi-fidelity oracles/surrogates are available in order to efficiently identify a diverse set of high-quality candidates within a given budget.

**Strengths:**

The multi-fidelity active learning scenario investigated in this work is of interest in various scientific discovery/design scenarios.
This work proposes how GFlowNet, a popular generative flow network model that can serve as an amortized sampler for drawing high-reward samples from a high-dimensional distribution, can be utilized for active learning under multi-fidelity setting.
According to a relatively simple and intuitive procedure outlined in Algorithm 1, this work shows that the proposed MF-GFN has the potential to identify a diverse set of high-scoring candidates at a lower acquisition cost compared to active learning schemes that rely on a single-fidelity acquisition function.


**Weaknesses:**

Although the proposed approach MF-GFN is reasonable, there are several major concerns regarding the current study.

1. While the authors claim that the proposed MF-GFN outperforms single-fidelity active learning as well as other multi-fidelity AL schemes, the evaluation results presented in the current manuscript (e.g., Figure 1, 2, 3) are not yet very convincing.
It appears that MF-GFN doesn't necessarily outperform other alternatives in a consistent manner, and when it does, the performance gain doesn't seem to be very significant.


2. For single-fidelity AL, the authors only consider the use highest fidelity oracle, which quickly consumes the AL budget.
Unless the high-fidelity samples lead to substantial learning improvement, it would be more desirable to use low-fidelity samples.
Of course, the actual relative value of high-fidelity vs. low-fidelity samples (considering the acquisition cost) would be different case-by-case, hence it is unclear whether the current examples provide fair comparison between MF vs. SF active learning.
To be fair, single-fidelity active learning performance should be evaluated at each of the considered fidelities to provide a more comprehensive picture of how SF AL would work at different fidelities.


3. On Hartmann function, MF-PPO clearly outperforms MF-GFN significantly. What are the characteristics of the Hartmann function that may lead to this discrepancy unlike some other examples considered in this work?


4. Comparisons across different examples should be more consistent.
Currently, different sets of methods are evaluated in different examples, and different K values were used for evaluating the top K samples.
This looks quite arbitrary and unless there is a clear reason for these choices, the same set of methods should be evaluated based on the different examples using the same K value (or same set of K values).


5. There should be further discussion on the computational cost of fitting h to D and retraining the GFlowNet in each iteration (of batch acquisition).
Considering that the multi-fidelity oracles may often be computational models with different computational cost, it may be sometimes (or often) more desirable to reduce training the GFlowNet multiple times and use this computational budget for a larger number of oracle evaluations.
As a result, this training cost should be considered in practice when designing and performing AL campaigns, and these practical aspects need to be discussed further.


6. There is currently no discussion regarding the impact of the batch size on the performance of MF-GFN and its comparison to other alternatives.


7.  Although (2) is a simple yet reasonable way of evaluating a potential sample considering both the acquisition cost and its value, it is not clear whether this would be a reliable estimate of the "value" of a given sample normalized by its acquisition cost.
There should be a better justification for this cost-adjusted utility function or at least some empirical evaluation.



**Questions:**

Please see the comments above in "Weaknesses".


**Limitations:**

The manuscript briefly discusses some potential limitations of the current study and its broader impacts.

---

> ### Author Rebuttal · Authors · 2023-08-08
>
> Dear Reviewer 3rWi,
>
> Thank you for the insightful review. We particularly appreciate the accurate summary, highlighting the specific challenges of scientific discovery that our work tackles, as well as the fact that you identified the most relevant strengths of our submission.
>
> Regarding the limitations highlighted in your review, let us address them one by one below.
>
> ### 1. “Evaluation results are not very convincing”
>
> You indicate in the review that “MF-GFN doesn't necessarily outperform other alternatives in a consistent manner, and when it does, the performance gain doesn't seem to be very significant.” Let us first re-emphasise that our goal was to design an active learning algorithm for scientific discovery to identify “novel, diverse, high-quality solutions”, as you rightfully mention in your review. Therefore, it is important to follow a wholistic approach to analyse the results. Namely, both diversity and high scores are required and good performance in only one of these metrics is not enough for the family of scientific discovery applications that are the target of our work.
>
> With this in mind, we note that MF-GFN achieves good performance in terms of both diversity and mean top-K scores in all the tasks evaluated. Crucially, our results show that MF-GFN (as well as the rest of GFN-based methods) are able to discover diverse candidates in all cases. Further, according to our results, MF-GFN significantly outperforms the alternative methods in most tasks. For example, on the AMP task, MF-GFN achieves the best mean energy (0.8) with 10x less budget than the next best methods. On the molecular ionisation potential (IP) task, MF-GFN achieves top mean energy over 100 samples  about -4.5) with ~40 % of the budget, while the next best method (MF-PPO) only achieves -6.5 mean energy with the same budget and only achieves its maximum (slightly better than MF-GFN) with 60 % of the budget. Importantly, our results reveal that while MF-PPO is able to find high scores (unsurprisingly, as a RL-based method) the set of top-K candidates has very low diversity in all cases (5-8 times less diverse than GFN-based methods).
>
> Finally, we would like to note that the improvements in budget utilisation achieved by MF-GFN can have a significant impact if translated to scientific discovery applications with very costly oracles, as is our plan in future work. By way of illustration, the budget savings displayed by MF-GFN in the molecular tasks with respect to SF-GFN, GFN with random fidelities and random sampling (we deliberately exclude MF-PPO in the comparison because of the lack of diversity) imply for example that many more molecules could be screened by the oracles with the same budget.
>
> ### 2. Single fidelity experiments with lower fidelity oracles
>
> We agree that this is an interesting comparison. Therefore, we have trained the single fidelity active learning methods in the molecular IP task (since it uses realistic oracles and costs) by using the lowest fidelity oracle. The results are provided in the separate page with figures (Figure 4b). We can see that (as expected) the single-fidelity method with the lowest-fidelity oracle uses less computational budget to find high-scoring candidates. However, it achieves lower mean scores than MF-GFN and slightly lower than SF-GFN with the highest-fidelity oracle, not being to leverage the available budget.
>
> ### 3. Why MF-PPO outperforms MF-GFN on the Hartmann task
>
> In the context of the Hartmann experiment, GFN initiates its exploration from the origin point. Conversely, the PPO commences from a random starting point within a bounded range, allowing at most three units of displacement (maximum possible displacement is 10 units) along each of the six axes. We hypothesise that this additional advantage aids the PPO algorithm in expediting the discovery of modes within the optimization process.
>
> ### 4. Evaluations with different values of K
>
> The values of K were selected so that they were consistent with (larger than) the acquisition size (active learning batch size), which in turn was set to approximately reflect realistic batch sizes in practical settings of the corresponding tasks. However, it is true that we selected K=100 in most cases as we could have selected K=75,200 or other values. In order to show that the conclusions of our experiments are consistent for different values of K, we provide a new set of results in the figures page with alternative values of K (Figures 2 and 3).
>
> ### 5. Computational cost of fitting the surrogate and GFlowNet
>
> While the computational cost of training both the surrogate model and GFlowNet are not negligible, in the practical settings where we expect MF-GFN to be of high value, the cost and time is largely dominated by the oracle queries, especially the higher fidelity oracles. To give a notion of the orders of magnitude, training the surrogate and GFlowNet takes hours, while evaluating a single molecule or material with DFT can take several days, or even weeks in the case of wet-lab experiments.
>
> ### 6. Impact of the batch size on the performance of MF-GFN
>
> We agree this is an interesting analysis point so we've repeated the molecule IP task with different batch sizes and provide the results in Figure 4a. We notice that the reward curve becomes steeper with higher batch sizes.
>
> ### 7. Reliability of the MF-MES acquisition function as proxy for the value of a candidate
>
> The effectiveness of MF-MES as a cost-utility function in contrast to other established approaches has been examined in literature [1]. Further, we have implemented the GIBBON formulation of MF-MES due to its notably reduced computational burden compared to the traditional variant. Empirical evidence indicates that the GIBBON formulation consistently outperforms alternative existing methods in the domain of multi-fidelity optimization [2].
>
> [1] Takeno et al., arXiv: 1901.08275
>
> [2] Song et al., arXiv: 1811.00755

---

> > ### Comment · Reviewer_3rWi · 2023-08-17
> >
> > First of all, I would like to thank the authors for the detailed and thorough response to the review comments.
> > In fact, the authors' rebuttal has addressed some of the doubts/concerns I had regarding the manuscript and has alleviated the concerns regarding the rest.
> > As a result, I will be happy to raise the overall evaluation score accordingly.

---

> > > ### Author Response · Authors · 2023-08-17
> > > **Ackowledgement for engaging in the discussion**
> > >
> > > Thank you for carefully considering our rebuttal, re-assessing the score and engaging in the discussion! Please let us know if any further concerns remain, which we would be happy to discuss.

---

### Official Review · Reviewer_TG5X · 2023-07-06

**Soundness:** 2 fair
**Presentation:** 3 good
**Contribution:** 2 fair
**Rating:** 5
**Confidence:** 4

**Summary:**

This paper introduce an algorithm for multi-fidelity active learning with GFlowNets and demonstrate that the proposed algorithm outperforms the baseline methods.

**Strengths:**

The paper is well written and includes two synthetic benchmark tasks and four practically relevant tasks for extensive experiment analysis.

**Weaknesses:**

The novelty of the paper's contribution may be questioned as it appears to bear similarities to existing works such as BMFAL (Li et al., 2022) and D-MFDAL (Wu et al., 2021) with the exception of the GFlowNets component.

Regarding the experiments, several existing multi-fidelity active learning baselines are missing, including DMFAL (Li et al., 2020a), BMFAL_Random (Li et al., 2022a), BMFAL (Li et al., 2022), D-MFDAL (Wu et al., 2021), and MF-BALD (Gal et al., 2017). Furthermore, the variants of the proposed method, namely GFlowNet with random fidelities and GFlowNet with the highest fidelity, seem more akin to an ablation study rather than proper baseline comparisons.

Additionally, the evaluation metrics employed, such as the mean score and mean pairwise similarity, are specific to GFlowNet. To ensure fair comparisons, it is recommended that the author considers adopting the evaluation metrics used in previous multi-fidelity active learning papers.

**Questions:**

1. Where are the experiment results for mean pairwise similarity?

2. What is the number of samples selected at each fidelity level?

**Limitations:**

The limitations are included.

---

> ### Author Rebuttal · Authors · 2023-08-08
>
> Dear Reviewer TG5X,
>
> Thank you for the reviews, for highlighting some of the strengths of work and for suggesting avenues for improvement. Below, we attempt to address your questions and the weaknesses one by one.
>
> ### Limited novelty
>
> The novelty of our contribution seems to be biggest concern in your review. In particular, you point that our work bears "similarities to existing works such as [1, 2] with the exception of the GFlowNets component". The work in [1] is a relevant contribution in the recent literature on multi-fidelity methods and as such as an important source of inspiration to our work. However, we believe there are crucial differences between our submission and BMFAL. You note that one difference is the introduction of the GFlowNet component.
>
> In our opinion, this is a substantial difference since not only is GFlowNet an alternative sampling method, but also it enables the application of multi-fidelity active learning to explore highly structured and high-dimensional candidates in combinatorially large spaces. In contrast, the majority of the multi-fidelity BO literature has focused on lower-dimensional, continuous spaces. Furthermore, GFN introduces the advantage of discovering diverse candidates instead of simply finding the mode (as is the approach in the BMFAL and D-MFDAL). Altogether, this enables the successful application of multi-fidelity methods in certain scientific discovery applications in which there has been little to no multi-fidelity literature to the best of our knowledge. As a matter of fact, we argue that the experiments we provide in this paper, leveraging the availability of multiple oracles in tasks such as DNA aptamer, antimicrobial peptide and molecular design, is a novel contribution in and on itself.
>
> The goal of the family of problems tackled by [1] is to find efficient PDE solvers where we have access to solvers at different resolution (fidelities). The goal of MF-GFN is to discover diverse candidates with certain properties in a combinatorially large space, having access to oracles with multiple degrees of confidence (and costs).
>
> Similar arguments can go to distinguish the contributions by D-MFDAL [2]---we believe the year of publication of D-MFDAL is 2023 though. First, the family of problems tackled by D-MFDAL is also solving PDEs, rather than diverse, high-scoring candidate generation. Second, we believe that the main (important) contribution in D-MFDAL paper is a novel methodological framework to improve the training of deep surrogate models in multi-fidelity active learning. Our contribution is not on the surrogate model side, but rather in the introduction of a generative model (GFlowNet) to explore the search space and sample diverse candidates.
>
> ### Missing baselines
>
> We have discussed this in the general comments to all reviewers. To sum up, in this work we tackle the problem of generating diverse, high-scoring candidates from combinatorially large, high-dimensional and structured spaces. This is a substantial difference with respect to most previously proposed multi-fidelity methods, such as the ones you mention. Most works in the BO literature focus on optimizing low-to-mid-dimensional, continuous spaces. Examples are predicting fluid dynamics in a rectangular domain or other problems that involve solving PDEs. On the active learning side, most have focused on pool-based active learning, where the goal is to efficiently train a predictive model by efficiently annotating a pool of samples. This is as well remarkably different to our purposes. For these reasons, the methods that you suggest in your review are unfortunately not directly applicable to the scientific discovery problems we explore in our paper.
>
> ### Evaluation metrics
>
> The evaluation metrics must be closely linked to the desiderata of the application problems. Given the different nature of the problems tackled in much of the multi-fidelity literature with respect to our work, as we have discussed above, the evaluation metrics must also be different. In particular, since our goal is not to train an accurate model, the typical metrics in pool-based active learning methods are not applicable to our work. In BO, a common metric is the regret. However, since our goal is not the optimisation of an unknown function, then regret would not be an accurate measure of success. In the scientific discovery applications we address, the goal is to find a diverse set of high-scoring samples, and our metrics are inspired by previous work that has tackled similar problems [3, 4].
>
> ### Where are the experiment results for mean pairwise similarity?
>
> We have admittedly not made a clear connection between the wording used in Section 4.1, which presents the metrics, and the presentation of results in the rest of the section. Mean pairwise similarity is actually the metric of *diversity* we use in our paper and the diversity results are provided in each plot as the colour of the markers, according to the colour legend shown on top of each plot. We will definitely use your feedback to improve the clarity of this important aspect.
>
> ### What is the number of samples selected at each fidelity level?
>
> For three randomly initialised runs of the molecules IP task, we provide the number of samples selected at each fidelity level once the total budget is expended. Number of samples in increasing order of fidelity:
> Seed 1: 635, 431, 305; Seed 2: 628, 490, 365; Seed 3: 1034, 216, 45. We will add these statistics for the other tasks in the camera-ready version.
>
> [1] Batch Multi-Fidelity Active Learning with Budget Constraints, Li et al., arXiv: 2210.12704
>
> [2] Disentangled Multi-Fidelity Deep Bayesian Active Learning, Wu et al., arXiv: 2305.04392
>
> [3] Biological Sequence Design with GFlowNets, Jain et al., arXiv: 2203.04115
>
> [4] Sample Efficiency Matters: A Benchmark for Practical Molecular Optimization, Gao et al., arXiv: 2206.12411

---

> > ### Comment · Reviewer_TG5X · 2023-08-18
> >
> > I thank the authors for the responses. They did address some of my concerns. I am open to reconsidering my evaluation if the authors promise to enhance their literature review with accurate references to the pertinent existing multi-fidelity active learning research.

---

> > > ### Author Response · Authors · 2023-08-19
> > > **Brief answer about extending the multi-fidelity literature review**
> > >
> > > Thank you for following up on our rebuttal answer! The updated manuscript will definitely include an extended review of the multi-fidelity literature. In particular, in Section 2 Related Work - which already includes a review of relevant multi-fidelity methods - we will discuss additional previous work on this area, including BMFAL (Li et al., 2022), D-MFDBAL (Wu et al., 2023) and Gal et al. (2017). We will also further clarify the differences of our work with the existing literature, in line with what we have discussed in the rebuttal.
> > >
> > > Additionally, if you think any other relevant work is missing, we would be grateful to know. Finally, we remain open to discuss any other aspects that you may consider unresolved.
> > >
> > > Li et al. Batch Multi-Fidelity Active Learning with Budget Constraints. NeurIPS 2022.
> > >
> > > Wu et al. Disentangled Multi-Fidelity Deep Bayesian Active Learning.  ICML 2023.
> > >
> > > Gal et al. Deep Bayesian Active Learning with Image Data. ICML 2017.

---

### Official Review · Reviewer_qmsr · 2023-07-07

**Soundness:** 4 excellent
**Presentation:** 4 excellent
**Contribution:** 2 fair
**Rating:** 6
**Confidence:** 4

**Summary:**

The authors adapt the standard GFlowNet framework to include a fidelity measurement for the oracle, and demonstrate on synthetic, biological and chemical datasets that, in almost all cases, MF-GFN outperforms relevant baselines in terms of achieving sampling performance within a fixed budget.

**Strengths:**

### Originality

The paper applies multi-fidelity ideas from Bayesian Optimization to the GFN framework. This is the first time such a thing has been done, and the standard GFN framework needed to be updated to sample a fidelity as well.

### Quality

The quality of the paper and results is sufficient for publication. The synthetic benchmarks are standard for this area of research, and the biological examples for aptamers and peptides are biologically relevant. The QM results for small molecules are less relevant than other tasks (e.g. ADMET in drug discovery) could have been.

### Clarity

The paper is very well written and easy to understand.

### Significance

The paper is moderately significant since the fidelity configurations are contrived and simple and likely do not represent experimental drug discovery fidelities (see Weaknesses below).

**Weaknesses:**

- the multi-fidelity framework is rather simple and would apply in situations when the oracle is computational (e.g. running DFT) rather than experimental (e.g. running biochemical assay), since computation costs are easily assumed to be uniform and applicable per sample, whereas experimental costs are more complex, can require batch acquisition rather than single sample, and the results can be far noisier in general. While it is infeasible to perform such a study for this paper, a synthetic example could be constructed with such properties.
- In the main paper, each task is only run with a single multi-fidelity configuration, but it would be interesting to run the same task but with different MF configurations in order to understand how the distribution of fidelities effects convergence per fraction of budget spent. Appendix D.1 does this once for the aptamer example, but a more comprehensive study, perhaps on synthetic data, would be enormously instructive.

**Questions:**

- In most examples, it is not clear how, at each fidelity level, the accuracy of the oracle relates to the cost. E.g. in the aptamer example, is the accuracy of an oracle trained on 1m aptamers 100x worse than the free energy calculation of the secondary structure? What is the relationship in the other examples?
- Except for the PPO method, diversity seems to be pegged at its highest level in all examples. Is there a more nuanced discussion you could give about this diversity? Like number of modes?

**Limitations:**

There is a sufficient discussion of limitations in the paper.

---

> ### Author Rebuttal · Authors · 2023-08-08
>
> Dear Reviewer qmsr,
>
> Thank you for your insightful reviews. We appreciate your comments about the strengths and weaknesses of our work. Below, we address each of the weaknesses you pointed out as well as your questions.
>
> ### Non-uniform per-sample oracle costs
>
> This is an interesting and important point indeed, since it is true that in practice, in many scientific discovery applications the cost per sample is not uniform. However, to the best of our knowledge, there is little to no literature on multi-fidelity methods with non-uniform costs for scientific discovery and little literature on more general active learning and Bayesian optimisation. One early example of active learning with varying annotation costs is the work by Settles et al. (NeurIPS workshop, 2008). Given the early stage of the application of multi-fidelity methods in challenging and practically relevant scientific discovery methods, we humbly believe that our work provides a significant contribution to the field and we hope to incorporate varying costs in future work. As a preliminary comment, in principle, we can incorporate such costs within the MF-GFN framework by replacing the costs $\lambda_m$ with the function $\lambda(x,m)$ which defines the cost of evaluating each candidate with each oracle
>
> ### Experiments with multiple cost distributions
>
> We agree that this an interesting aspect of the analysis and as you indicate, we included a set of results in the appendix on the DNA task. The costs of the oracles used in the molecular tasks reflect the actual relative costs and trying different cost distributions would not only require significant computational demands, but also impact the practical relevance of the results. Nonetheless, in order to shed more light on this aspect, we are providing additional results on the synthetic Hartmann function in the separate figures page of this rebuttal. As you may see, the results are consistent with those provided for the DNA task.
>
> ### Relationship between oracle costs and accuracy
>
> Regarding the DNA and AMP tasks, we calibrated the cost differences between oracles by drawing from real-world scenarios where practical experiments, like wet lab experiments, might take hours for sequence evaluation, while online simulations might only require a few minutes (hence, a 100-fold magnitude difference). Furthermore, as the low fidelity oracles of the AMP experimental setting displayed similar explained variances, we assigned equivalent costs to them. In the molecules experiment, we employed commonly used quantum chemistry packages for molecular modelling as our oracles, and the costs were established based on average evaluation times across a batch of 1000 molecules. For the synthetic experiments, we relied on costs and oracles borrowed from existing literature [1, 2].
>
> ### A more nuanced discussion on diversity
>
> As you indicate, except PPO, the rest of the methods achieve good levels of diversity, which speaks well about the proposed method. The reason is that the rest of the methods are either random samplers, which naturally produce diverse objects, and GFlowNet-based methods, which have been repeatedly shown to sample diverse candidates, since this was one of the core design purposes of its conception, according to the original paper [3]. In case of doubt, the diversity scores *are not* invariant throughout the active learning rounds, but in fact decrease slightly for GFN- and random-based methods.
>
> As you suggest in your review, an alternative angle to analyse the diversity of the samples is looking at the number of modes. We are happy to note that this is precisely the angle of the results presented in Appendix D.2 Energy of Diverse Top-K. In particular, we here restrict the inclusion in the set of top K samples by a measure of similarity to the elements in the set and we then provide the mean score of this set of diverse top-K samples. In other words, this metric can be regarded as the mean score of the top-K modes found. We will provide additional details in this section in the camera-ready version of the paper.
>
>
> [1] A General Framework for Multi-fidelity Bayesian Optimization with Gaussian Processes, Song et al., arXiv: 1811.00755
>
> [2] Multi-Fidelity Bayesian Optimization via Deep Neural Networks, Li et al., arXiv: 2007.03117
>
> [3] Flow Network based Generative Models for Non-Iterative Diverse Candidate Generation, Bengio et al., arXiv: 2111.09266

---

> > ### Comment · Reviewer_qmsr · 2023-08-21
> >
> > I would like to thank the authors for their thoughtful rebuttal and for providing additional studies with synthetic data. Reading through the  other rebuttals has helped me understand the contributions of this paper better and its context within the related BO literature. I am willing to raise my rating to Accept.

---

> > > ### Author Response · Authors · 2023-08-21
> > > **Thank you for the review and open to further discussion**
> > >
> > > We sincerely thank you for carefully reviewing our rebuttal answers to both your own concerns as well as other reviewers’. We are glad to read that they have been helpful in clarifying these concerns and improving the understanding of our paper. We are working towards incorporating the insights from this discussion into the paper itself too. If other concerns remain, we will be happy to further discuss.

---

### Official Review · Reviewer_KJZ2 · 2023-07-21

**Soundness:** 3 good
**Presentation:** 3 good
**Contribution:** 2 fair
**Rating:** 4
**Confidence:** 4

**Summary:**

In this submission, the authors proposed to apply GFlowNets as a sampler to sample for active learning based on the selected acquisition functions, instead of directly optimizing them in the procedure of Multi-fidelity Bayesian Optimization (MFBO). Even though focusing on active learning applications, the authors presented the method more in the MFBO setting, which aims at optimizing a target function by iteratively querying it as well as several different low-fidelity low-cost surrogate functions. In this work, a multi-fidelity Gaussian Process was used as multi-fidelity surrogates and multi-fidelity MES was chosen as the acquisition function. The main focus of the submission is to adopt GFlowNets for MFBO to query according to the acquisition function and the authors claimed that it improves preferred diversity of queries samples.

The authors tested the performance of the proposed method with single fidelity BO with GFlowNet, random fidelity with GFlowNet, random selection, and Multi-fidelity PPO on synthetic Branin and Hartmann functions as well as real-world tasks on DNA Aptamers, protein design, and small molecule design. Although the proposed method does not always achieve the best performance, the authors claim that it has better sample efficiency with comparable score optimization performance.

**Strengths:**

GFlowNet was implemented for MFBO, specially for active learning tasks. The authors tested the performance of the proposed method with single fidelity BO with GFlowNet, random fidelity with GFlowNet, random selection, and Multi-fidelity PPO on synthetic Branin and Hartmann functions as well as real-world tasks on DNA Aptamers, protein design, and small molecule design. Although the proposed method does not always achieve the best performance, the authors claim that it has better sample efficiency with comparable score optimization performance.

**Weaknesses:**

1. The main concern is that the methodological contribution is limited. The authors are mostly using the existing acquisition functions as the reward function in GFlowNet to solve multi-fidelity active learning. There is not much theoretical analysis of this GFlowNet-based active learning strategy throughout the submission. A more serious concern is that the submission is very much similar as [1], by considering multi-fidelity settings but the fidelity was simply considered as an additional input variable. The whole pipeline and all the methods are very much the same.

[1] Jain, Moksh, et al. "Biological sequence design with gflownets." International Conference on Machine Learning. PMLR, 2022.

2. The explanation of using GFlowNet can be improved. As described in the 194th line of the main text, the joint posterior distribution of the input $X$ and fidelity index $m$ was modeled but with the constraints that a fidelity $m>0$ of a trajectory must be selected once and only once, from any intermediate state. The authors may want to first define the DAG (Directed Acyclic graph) of this GFlowNet model, explicitly explain the allowable state transition, forward/backward policies, etc.

3. The design of the multi-fidelity kernel may need further explanations. Especially, adding fidelity indices as additional input by adopting  $K_2(m_1, m_2)$ defined between lines 559 and 560 of Appendix does not seem to capture the difference of $m_1$ and $m_2$ and not really invariant when permuting the fidelity indices. How can this guarantee to select appropriate fidelity to query? The authors may want to discuss the intuition behind this design. Also the reference 68 does not exist in Appendix or main text.

4. The provided code does not seem to have GFLowNet-based implementation but only has random and PPO implementations.

5. Since the diversity of the queried data was advertised as a reason for utilizing GFlowNet, the authors may need to provide such validation results in the main text, for example by explicitly comparing the diversity at the end by different competing methods for the tasks. Also it may be better to provide more information on the PPO setup, for example whether it selects one sample or a batch of samples, because the `diversity' should be automatically taken care of by the acquisition function in the sequential setting and it may be tricky to select a batch of samples in this case.

6. The author used the Top-K score of K candidates in each active learning round. The hyper-parameter K, and other hyper-parameters in GP kernels, could be influencing the results and conclusions. Sensitivity analysis should be provided.

7. If GFlowNet-based sampling is the important contribution, then other MCMC sampling methods by acquisition functions besides PPO should be also benchmarked.

8. There are language errors, for example, 1) there are typos in the 148th and 149th lines of the main text; 2) in the 610th line of the Appendix: ‘where C is the cost if the highest fidelity oracle’ should be corrected as ‘where C is the cost of the highest fidelity oracle’; and many others.

**Questions:**


1. Many tested scenarios appear to be MFBO instead of active learning. Shouldn't the author provide some actual active learning experimental results?

2. How fidelity was taken care in this framework?

3. How sensitive the MFBO performance will depend on different hyperparameters?

4. How does the proposed method compare with either optimization-based and other MCMC sampling based methods?

**Limitations:**

Both methodological limitations and societal impact were discussed.

---

> ### Author Rebuttal · Authors · 2023-08-08
>
> Dear Reviewer KJZ2,
>
> Thank you for your insightful feedback. Regarding the missing GFlowNet code in the submission, this is indeed a mistake which we will fix, as the GFlowNet code is actually based on open-sourced implementations. In what follows, we address each of the weaknesses you mention in your review and answer your questions.
>
> ### Limited contribution
>
> In your review, you mention that our submission is very similar to [1]. This paper is definitely a source of inspiration for our work but we would like to argue that the extension of the applicability of GFlowNet (GFN) for multi-fidelity active learning is not a trivial contribution, since one could think of multiple options to do so and we have proposed one that is simple yet effective. Our methodological contribution by which we adapt the sampling mechanism to include the selection of the fidelity for each sample aims to strike a balance between the cost and the accuracy of the evaluation. The active learning setup used to discover diverse high reward samples has 4 main components: the surrogate model, acquisition function, the GFN, and the oracles. Notably, we've adapted each of these components to fit the multi-fidelity setting after thorough experimentation. Additionally, we have achieved promising outcomes in the context of small molecules experimentation-a setting that, to the best of our knowledge, has not previously been explored within the multi-fidelity framework. We strongly believe that our work makes a significant contribution to the multi-fidelity active learning literature, enabling settings which were previously intractable.
>
> ### Clearer explanation of the GFlowNet DAG
>
> Since the details of the GFlowNet design (including state transitions and policy models) are specific to each of the tasks, we included them in each the corresponding subsections of the tasks in Section 4 and Appendix B. In Section 3.3, we explained the details of the multi-fidelity GFlowNet that are general to all tasks.
>
> ### Multi-fidelity kernel
>
> The multi-fidelity kernel design utilized in this study has been sourced from [2] and we acknowledge the oversight in failing to reference this in our initial submission. They implement a Downsampling Kernel for the data fidelity parameter, in cases where it is relevant, along with an Exponential Decay Kernel for the iteration fidelity parameter, when applicable. As our experimental approach treats fidelity as akin to a data point, the implementation of the Downsampling Kernel has been incorporated.
>
> ### Diversity results
>
> We are not sure of having properly understood this concern (number 5). Diversity is indeed a core metric to evaluate the results of our method. In the unlikely case you missed this, the diversity results are included in each of the figures in the benchmark tasks, as the colour of the markers at each active learning iteration, according to the colour legend above each figure. As you can see, GFlowNet-based and random-sampling-based methods achieve good diversity scores, in contrast to PPO, as is expected.
>
> These diversity results and how they differ between methods are commented in the text. Is your question or suggestion about providing quantitative results in the text? As a matter of fact, in our updated current version of the manuscript, we have added quantitative results not only to the text, but also to the figures, by plotting a numerical scale in the colour maps (see figures attached).
>
> ### Details about PPO
>
> The PPO experiments are performed in the exact same setup as the MF-GFN, with the only difference being the learning algorithm for the policy. Specifically, the policy constructs a single candidate in a trajectory. To generate a batch of candidates we take samples from this policy. The policy is trained with the standard clipped PPO learning objective. Additionally, to improve the diversity and exploration in the PPO experiments we use a number of random initial steps (without which the policy gets stuck on a single candidate). This number is constrained by an upper limit set to one-third of the total length of the sample. It is intractable to have a batch of candidates generated by a policy since there is a combinatorially large number of candidates to select from. Consequently, we cannot rely on a acquisition function to provide the signal for diversity.
>
> ### Influence of K
>
> In order to shed more light on the influence of the choice of K for reporting the results of the top-K samples,  we have added results of varying the value of K. As you can see, the same conclusions largely apply for different choices of K.
>
> ### MCMC baselines
>
> In the growing literature of GFlowNet-based methods, MCMC has been used as a common baseline, revealing that, consistently, GFlowNet is more efficient at discovering multiple modes of the target distribution than MCMC, when the support is high-dimensional and very large. For these reasons, while we agree that the set of results would be more complete with MCMC baselines, we decided not to include them in the experimental setup [3, 4, 5].
>
> ### Active learning vs. Bayesian optimization
>
> As discussed in Section 2, we make use of surrogate models and acquisition functions typical of Bayesian optimisation, but we are not interested in "simply" optimising the unknown, target function, what connects our work with active learning. However, our work is also not akin to standard active learning, since our goal is not to "simply" learn an accurate predictive model, but rather discovering new, high-scoring, diverse examples.
>
> ### “How fidelity was taken care in this framework?”
>
> The fidelity is incorporated in all parts of the proposed framework. As detailed in Appendix A, the surrogate model and the acquisition function both account for the fidelity and as described in Section 3.3 the fidelity is a part of the GFlowNet action space.
>
> [1] arXiv: 2203.04115
>
> [2] arXiv: 1903.04703
>
> [3] arXiv: 2111.09266
>
> [4] arXiv: 2201.13259
>
> [5] arXiv: 2209.12782

---

> > ### Comment · Reviewer_KJZ2 · 2023-08-15
> > **active learning & fidelity**
> >
> > I thank the authors for the responses. After reading the responses and other reviews, I decided to keep my current score for the following reasons:
> >
> > 1. The presented work is more related to Bayesian optimization instead of active learning. The authors should have performed literature review and experiments in the context of Bayesian optimization for 'discovering new, high-scoring, diverse examples'. It is not clear either if that is the case, how significant is the new contribution compared to the previous paper on using GFlowNet for Bayesian optimization.
> >
> > 2. It is understandable to take fidelity as 'a part of the GFlowNet action space. However, the relationships between different fidelity models, accuracy as well as cost, were not taken care in the presented work. If that is indeed case, again, the claim of new contributions to 'multi-fidelity active learning' is not convincing.

---

> > > ### Author Response · Authors · 2023-08-16
> > > **Clarification on novelty and related work**
> > >
> > > Thank you for your response! We would like to clarify some details to address your concerns:
> > >
> > > 1. As we highlight in the paper in the paragraph starting L49 as well as our common response to all reviewers, the problems of scientific discovery we study are technically different from Bayesian optimization. As opposed to searching for a single candidate maximizing the value of a black-box function, we are interested in searching for diverse modes of the black-box function. Additionally, we are interested in the practically inspired scenarios where the search space is not continuous but rather discrete and structured (for example molecules). As we elaborate in Section 2, where we discuss literature from Bayesian optimization, active learning and active search, this is a setting which to the best of our knowledge - has not been studied in any prior work. We would be happy to add any work you think is missing.
> > >
> > > 2. It is not merely the fidelity being a part of the GFlowNet action space, but the fidelity being accounted for in the reward function as well. The acquisition function we use (reward for the GFlowNet), described in Appendix A, considers the mutual information between the value of the selected candidate at the selected fidelity with the maximum value of the highest fidelity oracle. This is also scaled by the cost of the oracle at the selected fidelity. In effect the reward of the GFlowNet accounts for the quality (accuracy) and cost of the different fidelities. Additionally in the Appendix we provide details about the effect of different fidelities on all tasks.
> > >
> > > We hope you will re-consider these aspects in your decision.

---

### Official Review · Reviewer_tpVK · 2023-07-25

**Soundness:** 4 excellent
**Presentation:** 2 fair
**Contribution:** 3 good
**Rating:** 7
**Confidence:** 3

**Summary:**

This paper offers a new framework for multi-fidelity active learning using Generative Flow Networks. Given the recent success of GFlowNets as models for sampling diverse candidates among terminating states in a DAG, the authors attempt to leverage this property to put a new spin on active learning, where instead of sampling from a pool of unlabeled candidates, new objects are sampled from the entire construction space. Furthermore, the GFlowNet is also responsible for sampling the fidelity at which to evaluate the object, which they later show can provide advantage over just sampling the objects themselves. Two important components of this framework are 1) the multi-fidelity GP proxy model, which is a surrogate for the true measure of 'goodness' of a sampled object, and 2) the multi-fidelity acquisition function, which is used as input to the reward function of the GFlowNet to encourage exploration of the construction space. The authors then justify this framework with results from a variety of domains showing that MF-GFN offers promising results in terms of its ability to effectively leverage the lower fidelity oracles to reduce the total cost of exploration compared with only using a single oracle.

**Strengths:**

The paper provides an original way of injecting the desirable properties of the generative flow network into the hot field of active learning, which is especially important for guiding modern research in being able to know what experiments to run next. The paper communicates the main ideas relatively clearly and effectively, and is not limited by restrictive assumptions. The paper also backs up their claims with experimental evidence from a variety of domains.

**Weaknesses:**

I felt like the biggest weakness of the paper was probably the lack of thorough results and as noted in the Limitations section, the lack of practical oracles. To be an effective framework for active learning, I think it would have been much more compelling to tackle some real problems that domain specific scientists are working on, instead of the synthetic (Branin and Hartmann) tasks where MF-PPO seems to do just as well as the proposed model.

Additionally, I felt that the paper was a bit rushed, as there were some glaring typos (e.g. line 303) and some important aspects that were not entirely clear, such as the actual reward function that was used by the GFlowNet. This was not made explicit until the supplementary material and made it difficult for me to understand how the acquisition function precisely tied into the rest of the framework.

The last thing that I think would be helpful for the reader to understand is to give details on the budget; things like how long does it take the GFlowNet to sample one object, how long does it take for the oracle to evaluate the object, etc. would be helpful for the reader to put the timescales of things into perspective.

**Questions:**

* One thing that was odd was the presence of the active learning round in the reward function (in the exponent of rho). For rho != 1, it seems like the GFlowNet would have to have the active learning round as input, otherwise it wouldn't know how to appropriately match the flows. However, I could not find this detailed in the paper, and so I was wondering if this could be clarified.

* What motivated the choice of using MES for the acquisition function? It felt like this ended up taking a lot of the paper to contextualize, and it seems like there are simpler choices (such as UCB or plain ES) that could have led to a more focused discussion of the MF-GFN itself.

* A nice feature of GFlowNets is the ability to train from partial trajectories. Because the goal of the paper is to reduce the overall cost of exploring a design space, I would be interested to hear if any consideration has been made to try to "early stop" some of the trajectories from the GFlowNet, for example if the estimated flow has gone below some threshold. It seems like this might be able to further reduce the cost of sampling. Or if this was not considered, was this due to the relatively cheap cost of generating samples from the GFlowNet compared with the relatively expensive evaluation of the oracle?

**Limitations:**

yes.

---

> ### Author Rebuttal · Authors · 2023-08-08
>
> Dear Reviewer tpVK,
>
> Thank you for your review. We appreciate the positive comments about our work as well as your helpful feedback to improve our paper. We have made sure to fix the typos and improve the clarity in the updated version. In the remainder of our rebuttal, we address each of your concerns.
>
> ### Breadth of experiments
>
> According to your review, the main concern lies in the paper's limited results and the absence of practical oracles. Additionally, you suggest that addressing challenges pursued by domain-specific scientists would have been more impactful than focusing on synthetic tasks like Branin and Hartmann.
>
> We would like to first note that our paper includes results on the commonly used Branin and Hartmann tasks just for the sake of completeness. The core of our experiments are the four other practically relevant tasks (Section 4.4), namely: DNA aptamer design, antimicrobial peptide design and small molecules generation. Second, our experiments involving small molecules were carried out using oracles that hold practical significance. This aligns with challenges actively pursued by domain-specific scientists. Discovering molecules with higher negative adiabatic ionization potential holds practical importance for applications like organometallic synthesis and the design of organic semiconductors. As detailed in Section 4.4.3, our MF-GFN method discovers desired molecules with just half the computational budget by the standard single oracle setup.
>
> As you note in your review, on small-scale tasks such as Branin and Hartmann, existing methods such as MF-PPO are effective enough and our proposed MF-GFN does not provide a substantial advantage. It is in tasks involving exploration of high-dimensional, very large spaces where MF-GFN provides significant improvements in terms of exploration efficiency and diversity, as shown by our results in Section 4.4. As an additional note, it is also relevant that the MF-PPO algorithm was run with an advantage over MF-GFN: We applied a “warm-up” start of the PPO algorithm with several random steps, because in experiments without that, it was unable to find the modes in the high dimensional space.
>
> ### Details about budget and time
>
> For the DNA and AMP tasks, designed as inexpensive prototypes of practically relevant (more expensive) experiments, the oracles are either pre-trained neural networks (for instance, see Table 4) or lightweight Python libraries, such as NUPACK (Appendix B.3) whose computational cost is not the bottleneck of the total training time. In contrast, the oracles used in the tasks with small molecules are quantum chemistry packages commonly used in molecular modelling (see Appendix B.5) whose computational cost is not negligible and is actually orders of magnitude more costly than sampling from the GFlowNet policy.
>
> Regarding the computational cost of sampling from a trained GFlowNet, it is nearly negligible compared to the oracle evaluations as we mention in the paper (Section 3.2). This is why we can afford to sample a large number N of samples to then select the best K, according to the acquisition function.
>
> ### Active learning round in the reward function
>
> The acquisition function (MES) exhibits increased sparsity as additional samples are discovered. In order to facilitate optimization, a linear reduction of the parameter β (with a scaling factor denoted by ρ) is implemented at each successive active learning round (hence the active learning round number is an input to the reward function) so as to scale up the rewards. Note that within an active learning round the GFlowNet samples from this fixed reward function and thus the policy need not be conditioned on the active learning round.
>
> ### Choice of MES as acquisition function
>
> We agree that it would be interesting to repeat the experiments with alternative acquisition functions, such as a multi-fidelity version of UCB [1], but since our main contribution is the introduction of GFlowNet as a generative model in the multi-fidelity active learning setting, experimenting with multiple acquisition functions was out of scope. The choice of a competitive acquisition function such as MES makes random sampling a stronger baseline, as well as more efficient than plain ES [2]. Secondly, we would like to note that recent multi-fidelity methods proposed in the literature [3, 4, 5] have adopted or adapted information theory-based acquisition functions, such as MES, rather than UCB and EI. Finally, it is worthwhile to mention that we have adopted the GIBBON formulation of MF-MES which has been experimentally shown to consistently outperform other existing methods in the context of multi-fidelity optimization.
>
> ### Train GFlowNet from partial trajectories
>
> We are unfortunately not sure to have understood this question. If you refer to GFlowNet loss function that are able to assign credit to partial trajectories (Pan et al., 2023), we agree that this will likely improve the training efficiency and future work should consider the use of such objectives.
>
> [1] Multi-fidelity Gaussian Process Bandit Optimisation, Kandasamy et al., arXiv: 1603.06288
>
> [2] Max-value Entropy Search for Efficient Bayesian Optimization, Wang and Jegelka, arXiv: 1703.01968
>
> [3] Disentangled Multi-Fidelity Deep Bayesian Active Learning, Wu et al., arXiv: 2305.04392
>
> [4] Deep Multi-Fidelity Active Learning of High-dimensional Outputs, Li et al., arXiv: 2012.00901
>
> [5] Batch Multi-Fidelity Active Learning with Budget Constraints, Li et al., arXiv: 2210.12704
>
> [6] A General Framework for Multi-fidelity Bayesian Optimization with Gaussian Processes, Song et al., arXiv: 1811.00755

---

> > ### Comment · Reviewer_tpVK · 2023-08-16
> >
> > Thank you for the in depth response! It has helped me better understand your work. I believe the my current score is fair as the paper does produce intriguing results with a novel active learning pipeline, but as other reviewers have pointed out, the extent of the novelty may be limited as there are many similarities to [1], albeit with a different acquisition function and fidelity as an output of the model. I also think having a complete code base with the GFlowNet implementation would further strengthen this paper.
> >
> > [1] Jain, Moksh, et al. "Biological sequence design with gflownets." International Conference on Machine Learning. PMLR, 2022.

---

> > > ### Author Response · Authors · 2023-08-17
> > > **Brief answers about novelty and code availability**
> > >
> > > Thank you for reading our rebuttal and engaging in the discussion.
> > >
> > > In your latest response, you mentioned as remaining concerns that the novelty with respect Jain et al. (2022) and the availability of a complete codebase including the GFlowNet implementation. Let us briefly answer these two points.
> > >
> > > Regarding the code, we can guarantee that the we will release a complete implementation, as it builds upon existing open source libraries and reproducibility and availability of our code is a core principle for us. We regret that, by mistake, we did not include the GFlowNet code in the submission.
> > >
> > > Regarding the differences with respect to the work by Jain et al. (2022), we would kindly point you to our [answer to Reviewer KJZ2](https://openreview.net/forum?id=2ZtGWNn37W&noteId=LLDo6GMOiE) under the section “Limited contribution”. In brief, while both papers deal with active learning with GFlowNets for scientific discovery, the multi-fidelity component has involved the adaptation of the four main parts of the algorithm: GFlowNet, surrogate models, acquisition function and oracles. Furthermore, we have provided an extensive empirical evaluation of our proposed multi-fidelity algorithm.
> > >
> > > We hope we have shed additional light on these aspects. If other concerns remain, we would be happy to discuss further.

---

### Author Rebuttal · Authors · 2023-08-08

We would like to thank all the reviewers for their constructive feedback about our paper. We are sure that the changes motivated by this feedback have improved the present manuscript and will also positively impact our future work.

We have responded to each reviewer individually, trying to address every concern and question. Here, we would like to emphasize a few points about concerns that are shared by more than one reviewer. We would also like to mention that we are including in the rebuttal a figures page with 1) a visual summary of the proposed algorithm and 2) results of additional experiments addressing main concerns and questions.

**Context**: The method that we present in this paper, multi-fidelity active learning with GFlowNets (MF-GFN), is motivated by a growing need in certain scientific discovery applications for efficient machine learning models that can effectively leverage the data and tools available for scientists. Specifically, in areas such as drug discovery and materials discovery, scientists have access to multiple tools that serve as proxies to characterize properties of potential new candidates. This motivates the need for multi-fidelity approaches which can operate on high-dimensional structures (such as the one we present here) as well as empirical results in science-related tasks.

**Absence of Bayesian Optimization and Active Learning Baselines**: While the fields of multi-fidelity Bayesian optimization and active learning have seen important progress in recent years, the specific characteristics of problems such as drug discovery and materials discovery are different from the settings typically targeted by most multi-fidelity methods, such as engineering or scientific problems involving, for example, finding solver parameters for solving partial differential equations, a common task found in this literature. These specific characteristics are that the search space is highly structured (for instance, small molecules, proteins or crystal structures) as opposed to optimization in a continuous Euclidean space as is typical in many PDE problems; high-dimensional and also combinatorially large (for example, the estimated search space for small molecules is about $10^{60}$). These properties limit the direct application of standard Bayesian optimization methods in such problems. Consequently, there are, to the best our knowledge, little to no previous multi-fidelity methods directly suitable for the tasks we tackle in this paper. For these reasons, we could not include multiple baselines for comparison, which is something that some of the reviewers would have liked to see. Instead, we constructed a baseline based on competitive reinforcement learning method (PPO), by using PPO as a multi-fidelity sampler instead of GFlowNet.

**************************************************On Diversity:************************************************** The scientific discovery tasks we tackle in this paper also have distinct goals, compared to typical tasks where multi-fidelity methods have been used in the literature. In particular, as opposed to methods where Bayesian optimization is the most suitable method, here the goal is not to find the optimum of an unknown target function, but rather find *multiple* diverse modes of the function. This differs also from the class of problems where pool-based active learning is typically used, where the goal is to train an accurate predictive model through an efficient annotation strategy. For these reasons, the metrics used to evaluate our method needed to be adapted from the metrics used in the BO and AL literature. Instead, we here used metrics proposed in the literature tackling similar scientific discovery problems [1, 2], such as the mean top-K score and the diversity of the top-K candidates.

**Comparison with MF-PPO**: Some reviewers mention that the MF-GFN does not always outperform MF-PPO and the other evaluated baselines. We would like to emphasize that because of the nature of these metrics and the desired properties they aim at measuring, both metrics (mean score and diversity of the batch) need to be assessed jointly. In other words, it is not enough to just have high mean score if the diversity is low, because that would be akin to having just one candidate with slight variations. Obviously, diversity by itself with low mean score is not only useless but trivial to obtain. With this in mind, we would like to note that MF-GFN achieves the best results out of all the methods evaluated when considering both metrics simultaneously. This is in stark constrast with PPO, which undoubtedly is able to find candidates with high scores in most tasks, but at the expense of diversity.

**Experiments on Synthetic Functions**: Some reviewers noted that MF-GFN does not excel in terms of performance in the synthetic Branin and Hartmann functions. These tasks were included in the evaluation for the sake of completeness, because these are tasks familiar to the multi-fidelity Bayesian optimization community. The goal with these tasks was to show that MF-GFN is also able to obtain results comparable to other multi-fidelity BO methods. However, the relative simplicity of these tasks does not require sophisticated methods. It is in high-dimensional, structured and very large spaces where MF-GFN provides the largest advantage, as we show in the evaluation on the remaining, science-related tasks (DNA, AMP and small molecules).

We hope these responses as well as the individual answers to each reviewer shed light on your concerns and questions, and we look forward to further discussion.

[1] Biological Sequence Design with GFlowNets, Jain et al., arXiv: 2203.04115;

[2] Sample Efficiency Matters: A Benchmark for Practical Molecular Optimization, Gao et al., arXiv: 2206.12411

---

### Decision · Program_Chairs · 2023-09-21

**Decision:**

Reject

**Comment:**

This paper proposes an approach for multi-fidelity active learning using GFlowNets with the goal of trading off accuracy for cost. Overall, reviewers appreciated the presentation of the work and the set of empirical results in the paper. The main concern of reviewers was the novelty of the contribution in comparison with several existing papers on GFlowNets and the Bayesian optimization literature. Another reviewer was concerned with the modeling of the fidelities and stated that the paper could be more clear in considering how different fidelity should be chosen based on kernels and costs. Finally, reviewers felt like the clarity of the exposition could be improved further and noted a number of typos in the current version of the manuscript. We encourage the authors to take the reviewers' comments into in consideration when revising this work for a future submission.